# Evaluation of the Antifungal Activity of Endophytic and Rhizospheric Bacteria against Grapevine Trunk Pathogens

**DOI:** 10.3390/microorganisms10102035

**Published:** 2022-10-14

**Authors:** Marcelo I. Bustamante, Karina Elfar, Akif Eskalen

**Affiliations:** Department of Plant Pathology, University of California, Davis, CA 95616, USA

**Keywords:** biocontrol, endophytes, rhizobacteria, grapevine trunk diseases

## Abstract

Grapevine trunk diseases (GTDs) are caused by multiple unrelated fungal pathogens, and their management remains difficult worldwide. Biocontrol is an attractive and sustainable strategy given the current need for a cleaner viticulture. In this study, twenty commercial vineyards were sampled across California to isolate endophytic and rhizospheric bacteria from different grapevine cultivars with the presence and absence of GTD symptoms. A collection of 1344 bacterial isolates were challenged in vitro against *Neofusicoccum parvum* and *Diplodia seriata*, from which a subset of 172 isolates exerted inhibition levels of mycelial growth over 40%. Bacterial isolates were identified as *Bacillus velezensis* (*n =* 154), *Pseudomonas* spp. (*n =* 12), *Serratia plymuthica* (*n =* 2) and others that were later excluded (*n =* 4). Representative isolates of *B. velezensis*, *P. chlororaphis,* and *S. plymuthica* were challenged against six other fungal pathogens responsible for GTDs. Mycelial inhibition levels were consistent across bacterial species, being slightly higher against slow-growing fungi than against Botryosphaeriaceae. Moreover, agar-diffusible metabolites of *B. velezensis* strongly inhibited the growth of *N. parvum* and *Eutypa lata*, at 1, 15, and 30% *v*/*v*. The agar-diffusible metabolites of *P. chlororaphis* and *S. plymuthica*, however, caused lower inhibition levels against both pathogens, but their volatile organic compounds showed antifungal activity against both pathogens. These results suggest that *B. velezensis*, *P. chlororaphis* and *S. plymuthica* constitute potential biocontrol agents (BCAs) against GTDs and their application in field conditions should be further evaluated.

## 1. Introduction

Grapevine (*Vitis vinifera* L.) is one of the most important crops worldwide due to the high commercial value of wine, raisins, and table grapes. The cultivated area contemplates Mediterranean and temperate climate regions, between latitudes 30° and 50°, gathering approximately 7.72 million hectares [1]. California is the largest grape producer in the United States, with 348,000 bearing hectares by 2019, of which 68.6% were destined for wine, 17.3% for raisins, and 14.0% for table grapes, altogether with a total value above USD 5.4 billion [2]. A wide range of pests and diseases may affect the crop; hence an intensive management program is often required, increasing production costs. Fungal diseases affecting the woody tissues, collectively known as grapevine trunk diseases (GTD), represent a major threat on a global scale [3]. Chronic infections result in poor or no development of vegetative structures after bud break due to a malfunction of the vascular system. Symptoms are diverse and progress over time, potentially resulting in collapse and eventually in the death of the entire plant. Consequently, vineyards show significant reductions in yield and lifespan, which elevates production costs and economic losses [3,4].

Botryosphaeria dieback, Eutypa dieback, Phomopsis dieback, esca and black foot are recognized as the most frequent and destructive GTDs. More than 133 unrelated fungal species have been reported to be causal agents, belonging mainly to the phylum Ascomycota and a few others to Basidiomycota [4]. Over the last three decades, the incidence of GTDs has increased significantly worldwide. The expansion of the grape cultivated area, the transition to high-density plantations, including trellis training systems, the adoption of mechanical pruning, and the banning of effective chemical fungicides (i.e., sodium arsenite, benomyl, carbendazim, and methyl bromide) have been discussed as contributing factors [4]. The fungal pathogens infect the grapevine primarily through pruning wounds [5,6,7], thus, control must include strategies to protect wounded tissues. Complete eradication is not possible; therefore, management must be focused on a multidisciplinary approach, including cultural practices and physical, biological, and chemical control strategies. In this regard, biocontrol has become increasingly attractive in viticulture, given the current trend of reducing the use of chemical pesticides due to their negative impact on the environment and workers’ safety [8,9].

The grapevine microbiome represents an important source of biocontrol agents (BCAs) since they play beneficial roles in plant fitness and health [10,11,12]. For instance, endophytic bacteria have the ability to enhance the grapevine tolerance to disease through different mechanisms, namely, by competition for nutrients and space, antibiosis, interrupting the pathogen signaling, or by inducing plant defenses [13,14]. Therefore, the concept of a “balanced microbiome” has recently gained notorious attention due to recent work on grapevine microbial communities and their impact in disease expression [15,16]. In this context, it has been shown that grapevines with a higher abundance of endophytic beneficial bacteria display less or no symptoms in vineyards with known history of GTDs [17]. On the other hand, the endophytic nature of latent infections caused by trunk pathogens pose an advantage for biocontrol treatments, allowing grapevines to strengthen their tolerance to biotic and abiotic stress before the switch from endophytic to pathogenic behavior [18,19]. However, there is still a lack of variety of BCAs available for growers and nursery managers to reduce the impact of GTDs [14,20]. Hence, this study aimed to identify and evaluate in vitro the potential of endophytic and rhizospheric bacteria obtained from commercial vineyards with the absence and presence of GTD symptoms located in the main grape-growing regions in California as BCAs against GTD-causing pathogens.

## 2. Materials and Methods

### 2.1. Sampling and Isolation of Bacterial Endophytes

Over the summer of 2019, twenty vineyards of wine, raisin, and table grapes were sampled across 10 counties in California (Figure 1). Eight vines were selected according to the presence (*n* = 3) and absence (*n* = 5) of externally visible GTD symptoms on each vineyard. Symptomatic vines showed cankers, dead arms, dieback, stunted shoots, or leaf tiger-stripes. Trunk, cordon, and root samples were collected from each grapevine using non-destructive methods [21]. Briefly, trunk and cordon samples were obtained by removing the bark (in an area of 10 cm^2^) with a sterile chisel and disinfecting the surface with ethanol 70%. Once dried out, the internal wood was drilled with sterile drill bits (6.35 mm diameter), and the sawdust was collected into Whirl-Pak bags (Nasco, Fort Atkinson, WI, USA). Root samples were obtained by digging the soil with a clean shovel approximately 15 cm away from the trunk (around irrigation line) and collecting feeder roots with adhered soil in clean plastic bags [22]. Samples were transported to the laboratory in coolers and isolations were carried out in 90 mm Petri dishes with half-strength nutrient agar (½ NA; BD Biosciences, San Jose, CA, USA). Wood fragments were spread evenly onto the medium using a sterile tweezer. Additionally, the remaining samples were incubated in phosphate-buffered saline (PBS) for seven days, and aliquots of 100 µL were plated on ½ NA plates. Feeder root samples were shaken vigorously to remove loose soil particles, and 1 g of roots with strongly adhered soil were mixed with 99 mL of sterile distilled water in Erlenmeyer flasks. The solution was homogenized in an orbital shaker at 120 rpm for 1 min and 100 µL aliquots of 1:10 and 1:100 dilutions were plated onto the ½ NA plates. Plates were incubated for 2 to 4 days at 26 °C in the dark and morphologically different colonies were transferred to fresh individual full-strength NA plates.

### 2.2. Initial Screening of Grapevine Endophytic and Rhizospheric Bacteria for Antifungal Activity against Neofusicoccum parvum and Diplodia seriata

Bacterial isolates (*n* = 1344) were initially screened in vitro against the mycelium of *N. parvum* and *D. seriata* through antagonism assays described previously [17]. Both fungal isolates (*N. parvum* UCD7395 and *D. seriata* UCD7767) were obtained from the fungal collection of the Eskalen laboratory at the Department of Plant Pathology, University of California, Davis, that were originally isolated from GTD-symptomatic grapevine samples. A 5 mm mycelial plug of each fungal pathogen was placed at the center of 90 mm Petri dishes with potato dextrose agar (PDA) and three bacterial isolates plus a control (sterile distilled water) were inoculated at equidistant points around the mycelial plug, i.e., 3 cm from the center, using a sterile toothpick that was previously introduced in fresh bacterial culture. Plates were incubated at 25 °C for 3–7 days until the mycelium reached the border of the plate toward the controls. The radius of fungal growth was measured with a digital caliper from the center to the edge of the colony towards each treatment. The percentage of mycelial inhibition was calculated for each isolate using Equation (1):Percentage of inhibition (%) = 100 [(*R* − *r*)/*R*], (1)
where *R* and *r* corresponded to the radii of fungal growth toward the control and toward the bacterial treatment, respectively [23]. The screening was initially performed using *N. parvum*, and bacterial isolates showing inhibition percentages above 40% were subsequently screened against *D. seriata* using the same methodology. Plates were prepared in triplicates and bacterial isolates that showed over 40% inhibition against the mycelium of both pathogens were selected for further analyses.

### 2.3. Molecular Identification of Bacterial Isolates

Selected isolates (*n =* 172) were cultivated in NA plates for 24–48 h to perform DNA extraction following the protocol provided by the Quick-DNA™ Fungal/Bacterial Miniprep Kit (Zymo Research Corp., Irvine, CA, USA). Amplifications of the 16S ribosomal DNA gene were carried out through PCR using the primers pair 27F/1492R [24]. For isolates belonging to the *Bacillus subtilis* species complex, fragments of the genes that encode for the gyrase subunit A (*gyrA*), RNA polymerase subunit B (*rpoB*), phosphoribosylaminoimidazolecarboxamide formyltransferase (*purH*), DNA polymerase III subunit alpha (*polC*), heat shock protein groEL (*groEL*) and the 16S rDNA were also amplified [25]. Given the large number of isolates associated with the *B. subtilis* species complex, seven representative isolates were sequenced for the five additional loci. PCRs were run in a T100™ thermocycler (Bio-Rad, Hercules, CA, USA). The reaction mixture consisted of 2 µL of template DNA (ca. 10 ng), 12.5 µL of GoTaq^®^ Green MasterMix 2X (Promega, Madison, WI, USA), 9.3 µL of nuclease-free water and 0.6 µL of each primer (10 µM), completing a total volume of 25 µL. PCR conditions for the 16S rDNA gene included a hot start of 5 min followed by 35 cycles of 1 min at 94 °C for denaturation, 1 min for primer annealing at 63 °C, and 2 min at 72 °C for primer elongation, and a final step of 10 min at 72 °C. For the *gyrA*, *rpoB*, *polC*, *purH* and *groEL* amplifications, PCR conditions consisted of 35 cycles of 30 s at 94 °C, 30 s at 56 °C (*gyrA*), 52 °C (*rpoB*), 46 °C (*polC* and *groEL*) or 50 °C (*purH*) for annealing, and 1 min at 72 °C for primer extension. PCR products were submitted for Sanger sequencing to Quintara Biosciences (Hayward, CA, USA). Raw sequences were assembled using Sequencher v5.4.6 (Gene Codes, Ann Arbor, MI, USA). Consensus sequences of each isolate were compared with the NCBI nucleotide database using BLAST (http://ncbi.nlm.nih.gov/Blast.cgi, accessed on 1 September 2022) to obtain a preliminary identification. Phylogenetic analyses were run using the 16S rDNA gene sequences of closely related species of preliminary identified isolates, and for isolates belonging to the *B. subtilis* species complex, a multi-locus approach was adopted. Alignments were carried out separately by locus using MAFFT v7 (https://mafft.cbrc.jp/alignment/server, accessed on 1 September 2022) [26] and included sequences of the selected bacterial isolates and closely related species (Appendix A). Alignments were depurated using Gblocks, selecting the less stringent options [27]. Concatenation of the loci utilized for the *B. subtilis* species complex (16S-*gyrA*-*rpoB*-*purH*-*polC*-*purH*) was performed manually by assembling the six alignments into one, using MEGA X [28]. Phylogeny was reconstructed using the maximum parsimony method and bootstrap test with 1000 replications in MEGA X.

### 2.4. Dual Antagonism Assays of Selected Bacterial Isolates against Grapevine Trunk Pathogens

Six representative isolates of *B. velezensis*, *Pseudomonas chlororaphis,* and *Serratia plymuthica* (Table 1) were individually challenged against the mycelium of *N. parvum* (UCD7395), *D. seriata* (UCD7767), *Lasiodiplodia theobromae* (UCD9051), *Eutypa lata* (UCD7746), *Diaporthe ampelina* (UCD7544), *Phaeoacremonium minimum* (UCD7770), *Fomitiporia polymorpha* (UCD7757), and *Ilyonectria liriodendri* (UCD7874). All these fungal pathogens were also obtained from the fungal collection of the Eskalen laboratory mentioned above. Due to the differential growth rate among these fungi, bacterial isolates were inoculated at different times. For fast-growing fungal pathogens (*N. parvum*, *D. seriata* and *L. theobromae*), the assay was carried out in 90 mm diameter Petri dishes with full-strength PDA, where both the pathogen and the bacterial isolate were inoculated simultaneously at 22.5 mm from the center of the plate in opposite directions. The remaining pathogens with a slower growth rate (*E. lata*, *D. ampelina*, *Pm. minimum*, *F. polymorpha*, and *I. liriodendri)* were inoculated in 55 mm diameter PDA plates for 48 h (96 h in the case of *F. polymorpha*) prior the bacterial isolate at 10 mm from the center in opposite ways. In both cases, the pathogens were inoculated by placing a 5 mm diameter agar plug with actively growing mycelium on one side of the plate, and the bacterial isolates were streaked as a line of approximately 30 mm on the opposite side with a sterile toothpick previously inoculated with fresh bacterial culture. The incubation period ranged between three to four days for fast-growing pathogens, and fourteen days for slow-growing pathogens. Evaluations of mycelial radii were carried out when the fungal colonies of the controls reached the border of the plate in the direction of the treatment and inhibition percentages were calculated as described in Section 2.2. Each plate was prepared in triplicate, and the experiment was performed twice.

### 2.5. Effect of Bacterial Agar-Diffusible Metabolites on Grapevine Trunk Pathogens

The six representative bacterial isolates were grown and fermented in LB broth for 7 days at 28 °C and 140 rpm in an orbital shaker (Incu-Shaker^TM^ 10L, Benchmark Scientific, Sayreville, NJ, USA). Diffusible metabolites were obtained by centrifugation at 5000 rcf for 10 min and filtration of the supernatant through 0.22 µm pore size filter units (Stericup^®^, MilliporeSigma, Burlington, MA, USA). Cell-free filtrates were added at increasing concentrations (1, 15 and 30% *v*/*v*) into PDA flasks when the media was approximately 50 °C after autoclaving. Control flasks did not receive bacterial filtrates. Fungal pathogens were inoculated at the center of Petri dishes containing the different treatments using 5 mm diameter plugs with actively growing mycelium. Evaluations of mycelial radii were carried out when the fungal colonies of the controls reached the border of the plate and inhibition percentage was calculated as described in Section 2.2. Each plate was prepared in triplicate, and the experiment was performed twice.

### 2.6. Effect of Bacterial Volatile Organic Compounds (VOCs) on Grapevine Trunk Pathogens

The six representative bacterial isolates were used to assess the effect of their volatile organic compounds (VOCs) against the mycelial growth of *N. parvum* and *E. lata*, representing a fast and a slow growing trunk pathogen, respectively. Bacterial isolates were inoculated onto 90 mm diameter Petri dishes with Luria-Bertani agar (LB, tryptone 10 g/L, sodium chloride 10 g/L, yeast extract 5 g/L, agar 18 g/L) using a sterile toothpick, streaking the entire surface of the agar. The fungal pathogens were inoculated at the center of Petri dishes with PDA using 5 mm plugs with actively growing mycelium. Both bottoms of each Petri dish were disposed against each other and sealed with a double layer of paraffin wax (Parafilm^TM^, Bemis Co. Inc., Neenah, WI, USA), placing the side inoculated with bacteria at the bottom. Control plates had no bacteria streaked onto the LB agar. Evaluations of mycelial radii were carried out when the fungal colonies of the controls reached the border of the plate and inhibition percentage was calculated as described in Section 2.2. Each plate was prepared in triplicate, and the experiment was performed twice.

### 2.7. Statistical Analyses

Percentages of inhibition were subjected to analysis of variance (ANOVA) using generalized linear models with the corresponding R packages in InfoStat v2008 (Houston, TX, USA). Normality and homoscedasticity were checked and corrected when necessary and means were separated using Fisher’s least significant difference test (*p* < 0.05). Data were plotted in GraphPad Prism v.5.03 (San Diego, CA, USA).

## 3. Results

### 3.1. Initial Screening of Grapevine Endophytic and Rhizospheric Bacteria for Antifungal Activity against Neofusicoccum parvum and Diplodia seriata

From the field sampling carried out in 20 commercial vineyards over the summer of 2019, a collection of 1344 endophytic and rhizospheric bacterial isolates was obtained and analyzed. The antagonism assays against the mycelium of *N. parvum* and *D. seriata* revealed that 172 isolates showed mycelial growth inhibition percentages over 40% against both pathogens. Phylogenetic trees indicated that 154 isolates (89.5%) corresponded to *B. velezensis* (Figure 2), whereas the remaining belong to a range of species of *Pseudomonas* (12 isolates, Figure 3), *S. plymuthica* (2 isolates, Figure 4) and other genera (4 isolates) that were excluded from this study. The 154 isolates of *B. velezensis* were preliminary analyzed using their 16S rDNA sequences alone, which clustered them altogether in a single clade with multiple species closely related to *B. velezensis* (data not shown). However, a six-locus data set (16S rDNA-*gyrA*-*rpoB*-*purH*-*polC*-*groEL*) allowed an accurate identification. Regarding their origin, B. velezensis isolates were obtained primarily from the woody tissues of asymptomatic vines, whereas *Pseudomonas* spp. and *S. plymuthica* isolates were mainly recovered from the rhizosphere of both symptomatic and asymptomatic vines (Figure 5).

### 3.2. Dual Antagonism Assays of Selected Bacterial Isolates against Grapevine Trunk Pathogens

The six selected bacterial isolates inhibited the mycelial growth of almost all the pathogens over the threshold (40%), except for *L. theobromae*, in which only half of the isolates reached inhibition levels above 40% (Figure 6). Differences (*p* < 0.05) on mycelial inhibition levels were detected among the bacterial isolates on each pathogen. On Botryosphaeriaceae species, bacterial isolates of the same species did not differ on inhibition percentages, except for *P. chlororaphis* isolates against *L. theobromae*, and isolates of *S. plymuthica* against *D. seriata*. Specifically, on *N. parvum*, inhibition levels were significantly higher with UCD10763 (*P. chlororaphis*) than with UCD10756 (*S. plymuthica*). On *D. seriata*, inhibition percentages were higher with UCD10631 (*B.*
*velezensis*) than UCD10757 (*P. chlororaphis*) and UCD10756 (*S. plymuthica*). On *L. theobromae*, both *Bacillus* isolates and UCD10763 (*P. chlororaphis*) caused higher inhibition levels than the remaining ones that did not reached the threshold of 40%. Then, on slow-growing fungal pathogens, more differences were observed among isolates of the same species. On *E. lata*, the highest inhibition levels were observed with UCD10614 (*B.*
*velezensis*) and UCD10763 (*P. chlororaphis*), followed by UCD10719 (*S. plymuthica*), UCD10631 (*B.*
*velezensis*) and UCD10757 (*P. chlororaphis*), and lastly, UCD10756 (*S. plymuthica*). On *D. ampelina*, only UCD10763 (*P. chlororaphis*) was significantly higher than UCD10757 (*P. chlororaphis*). On *Pm. minimum*, inhibition levels were significantly higher with UCD10719 (*S. plymuthica*), followed by both *B.*
*velezensis* isolates, and UCD10763 (*P. chlororaphis*) ranking third, and later UCD10756 (*S. plymuthica*) and UCD10757 (*P. chlororaphis*) ranking fourth. On *F. polymorpha*, the highest inhibitions were caused by both *P. chlororaphis* isolates, followed by *B.*
*velezensis*, and later *S. plymuthica* ranking third. However, no differences were observed with UCD10631 (*B.*
*velezensis*) and *S. plymuthica* isolates. Lastly, on *I. liriodendri*, UCD10719 (*S. plymuthica*) and UCD10763 (*P. chlororaphis*) caused the highest inhibition levels, followed by UCD10757 (*P. chlororaphis*), UCD10756 (*S. plymuthica*) and UCD10614 (*B. velezensis*) that ranked second, and UCD10631 (*B.*
*velezensis*) ranking third.

### 3.3. Effect of Bacterial Agar-Diffusible Metabolites against Grapevine Trunk Pathogens

The cell-free filtrates from bacterial suspensions fermented for seven days significantly (*p* < 0.05) reduced the mycelial growth of both *N. parvum* and *E. lata* (Figure 7). For both pathogens, the reduction in mycelial growth was dependent on the interaction (*p* < 0.05) between the isolate and the concentration level of metabolites in the agar. Differences in mycelial growth were detected among the bacterial isolates at all tested concentrations. The metabolites produced by *B. velezensis* isolates reached inhibition levels significantly higher against both pathogens when compared to the filtrates from *P. chlororaphis* and *S. plymuthica*. Notably, at 1% only the *B. velezensis* metabolites caused inhibition levels above 50%. Further, the metabolites of UCD10719 (*S. plymuthica*) ranked second in the inhibition of both pathogens at 15% and 30%. Specifically, at 15%, the two *P. chlororaphis* filtrates ranked third and fourth at inhibiting both pathogens, whereas the isolate UCD10756 (*S. plymuthica*) was the less toxic. At 30%, the filtrate of UCD10763 (*P. chlororaphis*) ranked third against both pathogens, whereas UCD10756 (*S. plymuthica*) ranked fourth against *N. parvum* and second against *E. lata*, and UCD10757 (*P. chlororaphis*) was the less toxic.

### 3.4. Effect of Bacterial Volatile Organic Compounds on Grapevine Trunk Pathogens

The volatile organic compounds (VOCs) released by the six selected bacterial isolates caused lower inhibition levels on the mycelial growth of *N. parvum* than of *E. lata* (Figure 8). On *N. parvum*, the VOCs produced by isolate UCD10763 (*P. chlororaphis*) and both *S. plymuthica* isolates yielded a higher inhibition level than the remaining isolates, with inhibition levels from 12.3% to 15.9% in average. On the other hand, on *E. lata*, the VOCs from both *P. chlororaphis* isolates caused inhibition levels ranging from 64.3% to 70.9% in average, followed by UCD10719 (*S. plymuthica*) with an inhibition of 35.5%, significantly superior to the rest of the isolates.

## 4. Discussion

This study shows that isolates of *B. velezensis*, *P. chlororaphis,* and *S. plymuthica* obtained from GTD-symptomatic and asymptomatic grapevines have inhibitory activity against eight common fungal pathogens responsible for Botryosphaeria dieback, Eutypa dieback, Phomopsis dieback, esca, and black foot in California. Previously, other species of *Bacillus*, *Pseudomonas* and *Serratia* have been investigated for their potential as BCAs against grapevine trunk pathogens [3]. Among them, *Bacillus* spp. have been the most studied in both laboratory and field settings [13,29,30,31,32]. Less frequently, different species of *Pseudomonas*, *Serratia*, *Paenibacillus*, *Pantoea*, *Paraburkholderia*, and *Streptomyces* have also been tested in laboratory and greenhouse trials [30,33,34,35,36,37,38]. Coincidently, this study revealed that from a subset of 172 isolates with potential biocontrol activity against GTD pathogens, the majority (89.5%) corresponded to *B. velezensis*, with a smaller proportion of *Pseudomonas* spp. (6.7%) and *S. plymuthica* (1.2%).

Selected isolates of *B. velezensis*, *P. chlororaphis* and *S. plymuthica* showed the antifungal effect when challenged directly against the pathogens and indirectly through the use of their agar-diffusible and/or volatile metabolites in vitro. Specifically, *B. velezensis* isolates showed inhibition levels above 50% against all the pathogens (except on *I. liriodendri*, with 43% of inhibition in average) by both direct confrontation and their agar-diffusible metabolites at 1, 15 and 30% *v*/*v*. On the other hand, *P. chlororaphis* and *S. plymuthica* isolates inhibited all the pathogens by direct confrontation similarly to *B. velezensis*, with levels above 40% (except on *L. theobromae*, with lower levels that ranged from 31.7 to 58% of inhibition) with some differences in a few fungal pathogens. However, their agar-diffusible metabolites were not as inhibitory as the ones produced by both *B. velezensis* isolates, where concentrations above 15% *v*/*v* were needed to reach inhibition levels over 40% against *N. parvum* and *E. lata*. When comparing isolates of *P. chlororaphis*, the metabolites produced by isolate UCD10763 were more toxic at 15% and 30% *v*/*v* against *E. lata*, and at 30% against *N. parvum* when compared to isolate UCD10757. Similar observations were found between *S. plymuthica* isolates, where UCD10719 metabolites caused higher inhibition levels at 15% and 30% *v*/*v* against *N. parvum*, and 15% *v*/*v* against *E. lata*, compared to isolate UCD10756. These results highlight the importance of selecting the proper bacterial isolates that exhibit higher antifungal effects and that these could be harnessed by treating grapevines with living bacterial inoculants and/or their extracted secondary metabolites. *Bacillus*, *Pseudomonas,* and *Serratia* species secrete a diverse range of secondary metabolites that are highly inhibitory against plant pathogens. For example, *B. velezensis* secretes antibiotics such as bacillopeptines, macrolactins, bacillaene, difficidin, amylolysin, bacilysin, lantipeptides and microcins [39], cell-wall degrading enzymes such as chitinase, protease and β-1,3-glucanase [40], antimicrobial polypeptides such as iturins, fengycins, and surfactins [41], and siderophores such as bacillibactin [42]. *P. chlororaphis* produce antibiotics such as phenazine, pyrrolnitrine, 2-hexyl 5-propyl resorcinol and hydrogen cyanide, and siderophores such as pyoverdine and achromobactine [43]. *S. plymuthica* synthesizes antibiotics such as haterumalides, prodigiosin and pyrrolnitrin, and lytic enzymes such as chitinases and glucanases [44,45]. Our results suggest a possible implication between these bacterial-derived metabolites and the antifungal activity observed against GTD-associated pathogens. Therefore, understanding the chemical diversity of these metabolites may help to understand their interactions with the physiology of the plant host and the pathogen, as well as improve processes associated with a BCA formulation such as extraction, and purification, among others.

The effect of VOCs produced by selected bacterial isolates on the mycelial growth of *N. parvum* and *E. lata* was also studied in order to elucidate other potential mechanisms of inhibition. VOCs have many functions as signaling molecules and, among them, they can have antifungal properties against different plant pathogens [43]. Our results showed that only *P. chlororaphis* and *S. plymuthica* VOCs caused a significant inhibition against *E. lata* and not against *N. parvum*. An explanation for this is that *N. parvum* has a higher growth rate and therefore did not allow any of the six bacterial isolates to produce sufficient VOCs to significantly reduce the mycelial development. Another explanation could be that *N. parvum* is not sensitive or highly tolerant to these molecules. Some of the VOCs produced by these bacterial species include 3-methyl-1-butanol and methanethiol in the case of *P. chlororaphis* [43] and sodorifen, alcohols, ketones, pyrazine, and sulfur compounds, in *S. plymuthica* [46,47]. Interestingly, VOCs produced by *B. velezensis* isolates did not arrest the mycelial growth of neither of the pathogens, which could also be explained by a lack of sensitivity by both fungal species, or insufficient time for toxic VOCs to be produced, or even the medium composition was not suitable for VOCs production. *B. velezensis* produce diacetyl, benzaldehyde and isoamyl alcohol, which are known to be toxic VOCs to different plant pathogens such as *Botrytis cinerea*, *Penicillium italicum* and *Monilinia fructicola* [48]. Nevertheless, some of these molecules can also activate plant defense responses [49,50], thus representing an indirect mechanism of action against GTD-associated pathogens.

We aimed to investigate the effect of selected bacterial isolates on a broad range of fungal pathogens responsible for GTDs in California. Previous studies have mainly focused on a few species, such as *E. lata* alone [29,30,51,52] or a Botryosphaeriaceae species, usually *N. parvum* [13,34,37,53,54]. Several others included two or three species associated with esca, Eutypa dieback and/or Botryosphaeria dieback [31,35,55,56,57,58]. Some groups have focused on black foot pathogens [36,59,60], but not many have contemplated multiple species representing more than three GTDs elsewhere [32,60,61]. Given the high diversity of causal agents involved with these diseases, it is critical to decipher the breadth of responses of multiple pathogens to the presence of a BCA and/or its metabolites, which will ultimately determine its effectiveness. For example, the inhibition levels on slow-growing pathogens (e.g., *E. lata*, *D. ampelina*, *Pm. minimum*, *F. polymorpha*) were higher than on fast growing fungi (e.g., Botryosphaeriaceae). Additionally, even between Botryosphaeriaceae, the inhibition percentages were higher on *N. parvum* and *D. seriata* than on *L. theobromae*. A longer exposure to the presence of the bacterium and its metabolites during fungal growth may explain these observations. This information allows to imply that timing of the application of BCAs as a preventative strategy is critical in suppressing the pathogen development.

Finally, our findings in this study revealed that the grapevine woody tissues, the rhizosphere, and the vineyard soil constitute a robust source of potential BCAs against GTDs. Our selected bacterial isolates, especially the ones identified as *B. velezensis* and *P. chlororaphis*, exhibited high levels of inhibition against eight fungal pathogens responsible for GTDs and their agar-diffusible and volatile metabolites demonstrated to be involved in the suppression mechanism. Therefore, these isolates alone or in combination could provide a broader spectrum of protection to grapevines against the development of GTD-associated symptoms. Since these isolates are natural inhabitants of grapevines, they are likely to be well adapted to their plant host [62]. *B. velezensis*, *P. chlororaphis*, and *S. plymuthica* are ubiquitous inhabitants of the soil, water bodies, plant roots, and fermented foods, and have been extensively studied elsewhere for their antagonistic activity against several fungal plant pathogens and plant growth promotion capability [63,64,65,66]. Antibiosis, lytic enzymes and siderophores are the most described mechanisms by which these bacterial species exert their beneficial effects on several plant hosts [67,68,69]. Furthermore, *B. velezensis* and *P. chlororaphis* are known to form biofilms on plant structures, which contribute to the protection of both the plant and the bacteria from dehydration, salinity, and nutrient deficiency, especially nitrogen [68,69]. Currently, we are evaluating selected isolates of *B. velezensis*, *P. chlororaphis*, and *S. plymuthica* on field trials for their prevention and curative abilities against common GTDs pathogens. Result from these field studies will help to develop commercially available BCA for the management of GTDs.

## Figures and Tables

**Figure 1 microorganisms-10-02035-f001:**
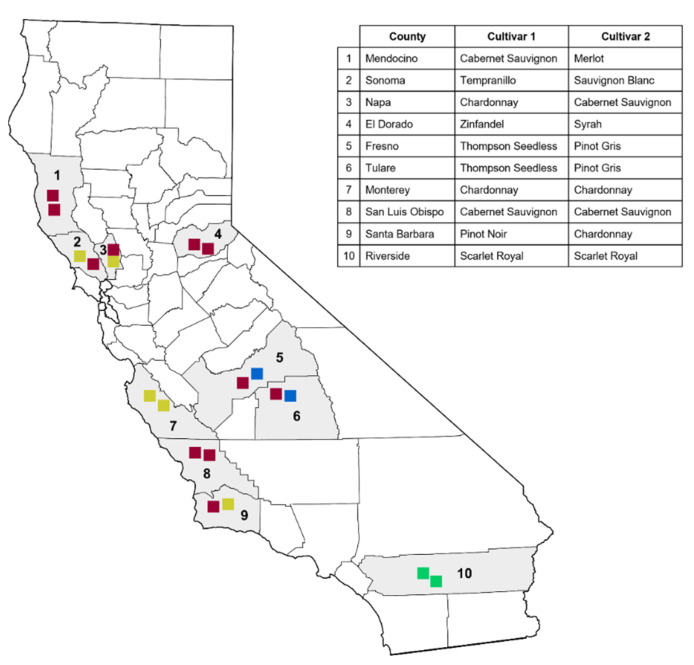
Sampled vineyards across California showing counties, type of vineyard (in colored squares) and cultivars. Red squares = red wine, yellow squares = white wine, blue squares = raisin, green squares = table grape.

**Figure 2 microorganisms-10-02035-f002:**
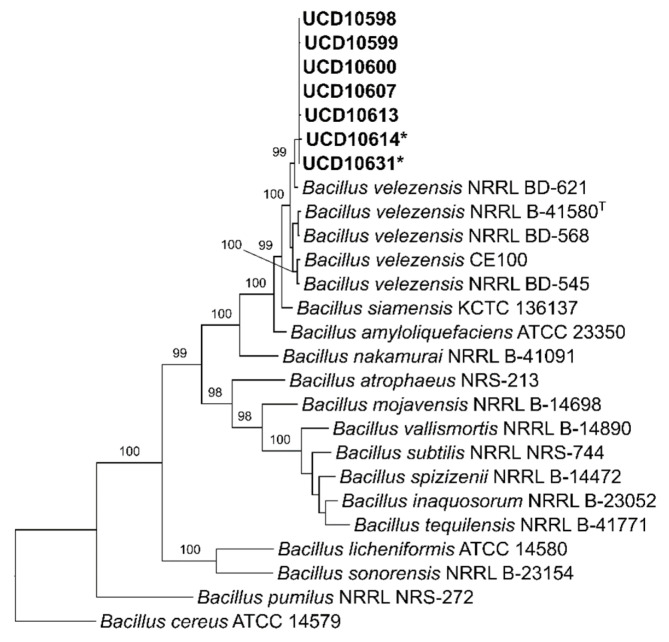
Most parsimonious phylogenetic analysis of seven isolates of *Bacillus velezensis* recovered from commercially grown various grapevine cultivars in California compared to closely related strains and species. The tree was inferred from a six-locus data set (16S rDNA-*gyrA*-*rpoB*-*purH*-polC-*groEL*). Numbers above branches represent non-parametric bootstrap values from 1000 replicates. *B. cereus* (ATCC 14579) was used as outgroup. ^T^ = type strain of *B. velezensis* (NRRL B-41580 = CR-502). * = isolates used in dual antagonism assays and metabolites analyses.

**Figure 3 microorganisms-10-02035-f003:**
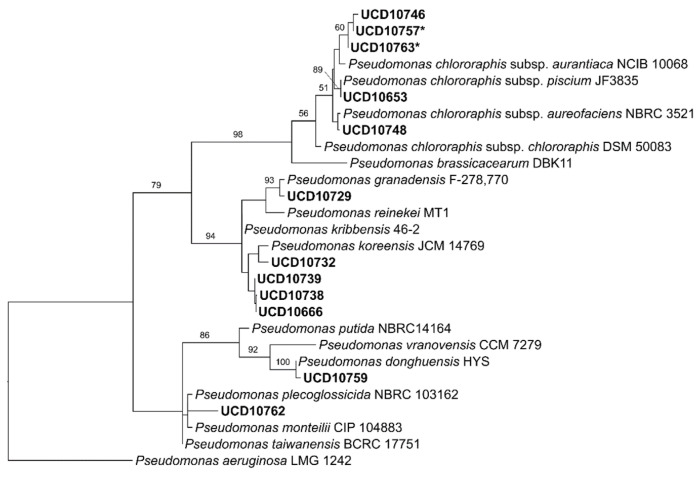
Most parsimonious phylogenetic analysis of 12 isolates of *Pseudomonas* spp. recovered from commercially grown various grapevine cultivars in California compared to closely related species. The tree was inferred with sequences of the 16S rDNA gene. Numbers above branches represent non-parametric bootstrap values from 1000 replicates. *P. aeruginosa* (LMG 1242) was used as outgroup. * = isolates used in dual antagonism assays and metabolites analyses.

**Figure 4 microorganisms-10-02035-f004:**
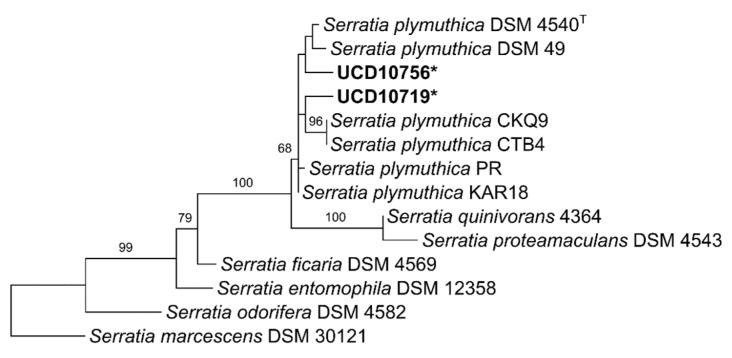
Most parsimonious phylogenetic analysis of two isolates of *Serratia plymuthica* obtained from commercial grapevines in California compared to closely related species. The tree was inferred with sequences of 16S rDNA gene. Numbers above branches represent non-parametric bootstrap values from 1000 replicates. *S. marcescens* (DSM 30121) was used as outgroup. ^T^ = type strain of *S. plymuthica* (DSM 4540 = K − 7). * = isolates used in dual antagonism assays and metabolites analyses.

**Figure 5 microorganisms-10-02035-f005:**
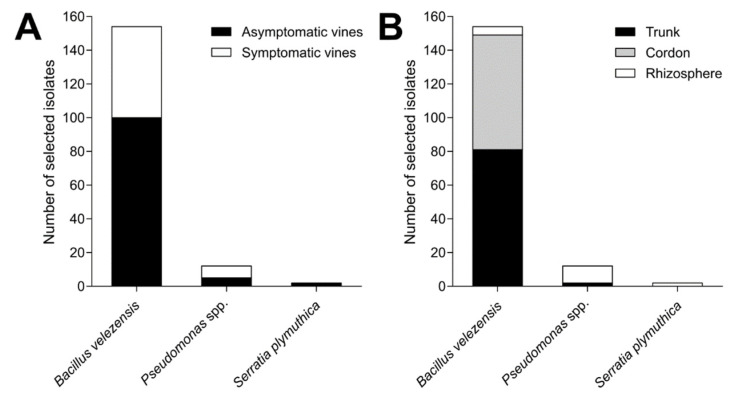
Distribution of selected bacterial isolates (*n* = 172) that showed inhibition levels over 40% against the mycelial growth of *Neofusicoccum parvum* and *Diplodia seriata* according to the vine health status (**A**) and the tissue they were recovered from (**B**).

**Figure 6 microorganisms-10-02035-f006:**
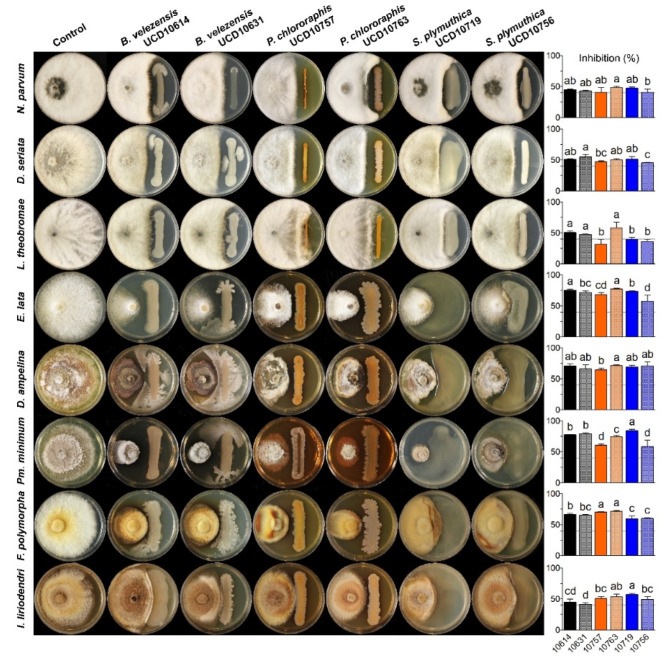
Inhibition levels (%) caused by selected isolates of *B. velezensis*, *P. chlororaphis* and *S. plymuthica* against *Neofusicoccum parvum*, *Diplodia seriata*, *Lasiodiplodia theobromae*, *Eutypa lata*, *Diaporthe ampelina*, *Phaeoacremonium minimum*, *Fomitiporia polymorpha*, and *Ilyonectria liriodendri*. Means with the same letter horizontally on each graph are not significantly different from each other according to the Fisher’s LSD test (*p* > 0.05). Gray line represents the threshold of 40% of inhibition.

**Figure 7 microorganisms-10-02035-f007:**
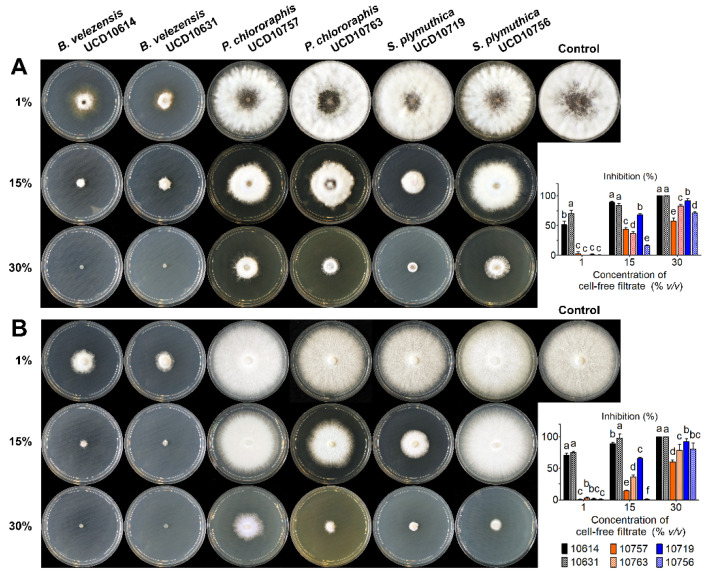
Inhibition levels (%) of increasing concentrations of agar-diffusible metabolites produced by selected bacterial isolates against the mycelial growth of *N. parvum* (**A**) and *E. lata* (**B**). On each graph, means with the same letter within each level of filtrate concentration are not significantly different from each other according to the Fisher’s LSD test (*p* > 0.05). The legend at the bottom right shows isolate codes for both graphs.

**Figure 8 microorganisms-10-02035-f008:**
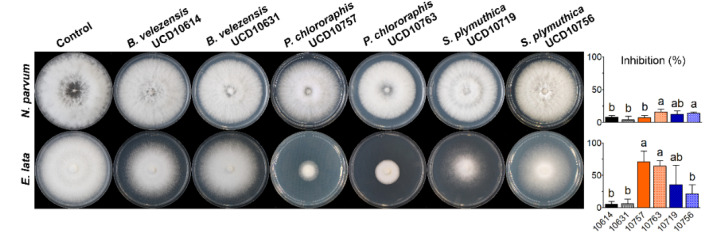
Inhibition levels (%) of volatile organic compounds (VOCs) produced by selected isolates of *B. velezensis*, *P. chlororaphis* and *S. plymuthica* against the mycelial growth of *N. parvum* (top) and *E. lata* (bottom). On each graph, means with the same letter horizontally are not significantly different from each other according to the Fisher’s LSD test (*p* > 0.05).

**Table 1 microorganisms-10-02035-t001:** Selected isolates of endophytic and rhizospheric bacteria obtained from commercial grapevines in California for in vitro antagonism and metabolite assays against GTD-causing pathogens.

Bacterial Species	Isolate	Tissue	Vine Health Status	County	Cultivar
*Bacillus velezensis*	UCD10614	Cordon	Asymptomatic	Santa Barbara	Pinot Noir
	UCD10631	Trunk	Asymptomatic	San Luis Obispo	Cabernet Sauvignon
*Pseudomonas chlororaphis*	UCD10757	Rhizosphere	Symptomatic	Monterey	Chardonnay
	UCD10763	Rhizosphere	Asymptomatic	Riverside	Scarlet Royal
*Serratia plymuthica*	UCD10719	Rhizosphere	Asymptomatic	Fresno	Thompson Seedless
	UCD10756	Rhizosphere	Asymptomatic	Tulare	Thompson Seedless

## Data Availability

Not applicable.

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
