# Peer review of "Evaluation of the Antifungal Activity of Endophytic and Rhizospheric Bacteria against Grapevine Trunk Pathogens"

_microorganisms, 2022, doi:10.3390/microorganisms10102035_

Round 1

Reviewer 1 Report

This research proposes to evaluate the efficiency of potential BCAs bacteria to control several pathogenic fungi associated with the grapevine trunk diseases.

The results, limited to in vitro assays evidenced the efficiency of 3 species to limit the growth of 8 GTDs associated fungi. In addition, the ability of extracellular and volatile compounds to limit the growth of 2 pathogenic fungi was tested, leading to hypothesize the mode of action of these potential BCAs.

The experiment is correctly carried out but some information about methods, number of replicates and statistical tests are missing. Plus, a larger review of the characteristics of the candidate BCAs should be performed in the discussion to offer a bigger hindsight on the topic.

L37-41: Add the citation supporting this affirmation

L69: It would be interesting to add the list of species already commercialized as BCAs to reduce GTDs

In the methods: the number of replicates for each test is not indicated

L164-173: precise the duration of the growth (test) for each species.

L170: Please check the size of the pathogen inoculated plug (45mm à 5mm?)

L173: the protocol for calculating the inhibition rate for this method is not described

L216: It should be clarified if the 147 other isolates not shown in figure 2 belong to the same cluster as the 7 shown in the figure.

L222: the affirmation should be supported by a statistical test

L274: The results shown in the photos do not always seem to fit with the histograms: for example the test concerning D. seriata. The strain 10756 seems to inhibit the growth of the fungi similarly to the other strains, so it seems strange to see the letter c on the histogram.  The lack of information on how the inhibition rate was calculated does not help to understand the results.

L328: indicate the inhibition rate to illustrate the adjective “strong”, ex : “inhibition level above….%”

L348: produces

Discussion: some bibliography about the characteristics of the 3 candidate BCAs could be added: where it is commonly isolated from (compare with this study), against which pathogens it is already used/commercialized?, the modes of action already identified (antibiosis/PGP…)

Serratia :

 Soenens, A., Imperial, J. Biocontrol capabilities of the genus Serratia. Phytochem Rev 19, 577–587 (2020). https://doi-org.scd-proxy.univ-brest.fr/10.1007/s11101-019-09657-5

Kshetri, L., Naseem, F., Pandey, P. (2019). Role of Serratia sp. as Biocontrol Agent and Plant Growth Stimulator, with Prospects of Biotic Stress Management in Plant. In: Sayyed, R. (eds) Plant Growth Promoting Rhizobacteria for Sustainable Stress Management . Microorganisms for Sustainability, vol 13. Springer, Singapore. https://doi-org.scd-proxy.univ-brest.fr/10.1007/978-981-13-6986-5_6

B. velezensis:

Alenezi, F.N.; Slama, H.B.; Bouket, A.C.; Cherif-Silini, H.; Silini, A.; Luptakova, L.; Nowakowska, J.A.; Oszako, T.; Belbahri, L. Bacillus velezensis: A Treasure House of Bioactive Compounds of Medicinal, Biocontrol and Environmental Importance. Forests 2021, 12, 1714. https://doi.org/10.3390/f12121714

Miao Ye, Xiangfang Tang, Ru Yang, Hongfu Zhang, Fangshu Li, Fangzheng Tao, Fei Li, and Zaigui Wang, Characteristics and Application of a Novel Species of Bacillus: Bacillus velezensis. ACS Chemical Biology 2018 13 (3), 500-505

DOI: 10.1021/acschembio.7b00874

Author Response

Dear reviewer,

Thank you for your time and your comments to improve our manuscript. We have accepted all your edits and the changes in the current version of the text are highlighted in blue.

Please find below our responses to your questions.

This research proposes to evaluate the efficiency of potential BCAs bacteria to control several pathogenic fungi associated with the grapevine trunk diseases.

The results, limited to in vitro assays evidenced the efficiency of 3 species to limit the growth of 8 GTDs associated fungi. In addition, the ability of extracellular and volatile compounds to limit the growth of 2 pathogenic fungi was tested, leading to hypothesize the mode of action of these potential BCAs.

The experiment is correctly carried out but some information about methods, number of replicates and statistical tests are missing. Plus, a larger review of the characteristics of the candidate BCAs should be performed in the discussion to offer a bigger hindsight on the topic.

L37-41: Add the citation supporting this affirmation

Response: Added.

L69: It would be interesting to add the list of species already commercialized as BCAs to reduce GTDs

Response: This list was referenced in L330, that corresponds to a thorough review in GTD management.

In the methods: the number of replicates for each test is not indicated

Response: Corrected in L118; L174-177; L189-191; L203-206.

L164-173: precise the duration of the growth (test) for each species.

Response: Corrected in L174-175.

L170: Please check the size of the pathogen inoculated plug (45mm à 5mm?)

Response: Corrected (5 mm diameter)

L173: the protocol for calculating the inhibition rate for this method is not described

Response: Corrected in L177-178.

L216: It should be clarified if the 147 other isolates not shown in figure 2 belong to the same cluster as the 7 shown in the figure.

Response: Clarified in L133-134 and L227-230.

L222: the affirmation should be supported by a statistical test

Response: The data of Figure 5 is descriptive and cannot be compared statistically since it represents frequency of isolates and not a measured variable with replicates.

L274: The results shown in the photos do not always seem to fit with the histograms: for example the test concerning D. seriata. The strain 10756 seems to inhibit the growth of the fungi similarly to the other strains, so it seems strange to see the letter c on the histogram.  The lack of information on how the inhibition rate was calculated does not help to understand the results.

Response: We understand the confusion that the pictures and the statistical analyses can create due to the small but significant differences between isolates. To clarify, for Diplodia seriata, the ANOVA showed a p=0.0076 and the following result:

Bacterial isolate

Mean percentage of inhibition of mycelial growth

Standard error

LSD test

B. velezensis UCD10631

55.07

1.47

a

S. plymuthica UCD10719

51.53

1.47

a b

B. velezensis UCD10614

50.93

1.47

a b

P. chlororaphis UCD10763

50.30

1.47

   b

P. chlororaphis UCD10757

47.07

1.47

   b c

S. plymuthica UCD10756

45.40

1.47

      c

The inhibition percentage calculation was added in L179. Maybe changing the scale of the graphs in Figure 6 may help to avoid confusions.

L328: indicate the inhibition rate to illustrate the adjective “strong”, ex : “inhibition level above….%”

Response: Corrected in L350, 351, 353, 354, and 355.

L348: produces

Response: The sentence is constructed in past tense. Therefore, ‘produced’ is grammatically correct. (Now in L356,359).

Discussion: some bibliography about the characteristics of the 3 candidate BCAs could be added: where it is commonly isolated from (compare with this study), against which pathogens it is already used/commercialized?, the modes of action already identified (antibiosis/PGP…). Serratia: Soenens, A., Imperial, J. Biocontrol capabilities of the genus Serratia. Phytochem Rev 19, 577–587 (2020). Doi: 10.1007/s11101-019-09657-5. Kshetri, L., Naseem, F., Pandey, P. (2019). Role of Serratia sp. as Biocontrol Agent and Plant Growth Stimulator, with Prospects of Biotic Stress Management in Plant. In: Sayyed, R. (eds) Plant Growth Promoting Rhizobacteria for Sustainable Stress Management. Microorganisms for Sustainability, vol 13. Springer, Singapore. Doi: 10.1007/978-981-13-6986-5_6. B. velezensis: Alenezi, F.N.; Slama, H.B.; Bouket, A.C.; Cherif-Silini, H.; Silini, A.; Luptakova, L.; Nowakowska, J.A.; Oszako, T.; Belbahri, L. Bacillus velezensis: A Treasure House of Bioactive Compounds of Medicinal, Biocontrol and Environmental Importance. Forests 2021, 12, 1714. Doi: 10.3390/f12121714. Miao Ye, Xiangfang Tang, Ru Yang, Hongfu Zhang, Fangshu Li, Fangzheng Tao, Fei Li, and Zaigui Wang, Characteristics and Application of a Novel Species of Bacillus: Bacillus velezensis. ACS Chemical Biology 2018 13 (3), 500-505. doi: 10.1021/acschembio.7b00874

Response: The discussion has included the suggested topics and added the corresponding references in L426-434.

Reviewer 2 Report

Article title:

Evaluation of the Antifungal Activity of Endophytic and Rhizospheric Bacteria against Grapevine Trunk Pathogens"

I suggest that this article be published based on research that sought to identify and assess in vitro the potential of endophytic and rhizospheric bacteria obtained from commercial vineyards with the absence and presence of grapevine trunk diseases symptoms located in the primary grape-growing regions of California.

 The work done is certainly of international interest and the format applied is certainly suitable for a research article. The work is original, of particular interest, and can certainly stimulate research on this topic. The length of the article is appropriate for the journal and the graphs and tables are clear and easy to understand. The conclusion summarizes the aims of the work and future prospects.

Just be sure to write the scientific names of microbes in italics, especially from lines 214-221, Figures 2, 3, and 4.

Author Response

Dear reviewer,

Thank you for your time and your comments to improve our manuscript. We have accepted all your edits and the changes in the current version of the text are highlighted in blue.

Please find below our responses to your questions.

I suggest that this article be published based on research that sought to identify and assess in vitro the potential of endophytic and rhizospheric bacteria obtained from commercial vineyards with the absence and presence of grapevine trunk diseases symptoms located in the primary grape-growing regions of California.

 The work done is certainly of international interest and the format applied is certainly suitable for a research article. The work is original, of particular interest, and can certainly stimulate research on this topic. The length of the article is appropriate for the journal and the graphs and tables are clear and easy to understand. The conclusion summarizes the aims of the work and future prospects.

Response: Thank you very much for your comment.

Just be sure to write the scientific names of microbes in italics, especially from lines 214-221, Figures 2, 3, and 4.

Response: Scientific names were and are currently in italics. Possibly the first manuscript suffered undesired changes when used in different versions of Microsoft Word. The current version is provided on pdf for verification.

Reviewer 3 Report

Manuscript ID: microorganisms-1979692

Evaluation of the Antifungal Activity of Endophytic and Rhizospheric Bacteria against Grapevine Trunk Pathogens

The The authors describe an extensively microbiological sampling strategy to identify bacterial strains with high potential in the inhibition of fungal vine pathogens. The results of the bacterial characterization led to the identification of six bacterial strains, from the following species: Bacillus velezensis, Pseudomonas chlororaphis, Serratia plymuthica, whit high toxicity and inhibitory profiles against multiple fungal pathogens. Observed inhibitions could be associated with the production of secondary and volatile metabolites by bacterial strains. The isolated strains have the potential for application as biocontrol agents in vine crops.

The manuscript could be considered for publication after minor corrections.

The following recommendations must be addressed before accepting the manuscript for publication.

Line 214, the names of the fungal species must be in italics

Lines 217-221, the names of the bacterial strains must be in italics

Line 229, the names of the bacterial strains must be in italics

Lines 232 and 2the names of the bacterial strainsames must be in italics

Lines 238, 240 and 241, the names of the bacterial strains must be in italics

In figure 4, why you did not use a more closely related bacterium species, such as Serratia marcescens or some similar species as an outgroup in phylogenetic analysis?

In the whole manuscript, there is a mixture of short and full names of the microorganisms, please review to make sure that the microorganism's names are in the full form the first time mentioned and then, only use the short name.

Author Response

Dear reviewer,

Thank you for your time and your comments to improve our manuscript. We have accepted all your edits and the changes in the current version of the text are highlighted in blue.

Please find below our responses to your questions.

The authors describe an extensively microbiological sampling strategy to identify bacterial strains with high potential in the inhibition of fungal vine pathogens. The results of the bacterial characterization led to the identification of six bacterial strains, from the following species: Bacillus velezensis, Pseudomonas chlororaphis, Serratia plymuthica, whit high toxicity and inhibitory profiles against multiple fungal pathogens. Observed inhibitions could be associated with the production of secondary and volatile metabolites by bacterial strains. The isolated strains have the potential for application as biocontrol agents in vine crops.

The manuscript could be considered for publication after minor corrections.

The following recommendations must be addressed before accepting the manuscript for publication.

Line 214, the names of the fungal species must be in italics

Lines 217-221, the names of the bacterial strains must be in italics

Line 229, the names of the bacterial strains must be in italics

Lines 232 and 2the names of the bacterial strains names must be in italics

Lines 238, 240 and 241, the names of the bacterial strains must be in italics

Response: Thank you for your comments. Scientific names were and are currently in italics. Possibly the first manuscript suffered undesired changes when used in different versions of Microsoft Word. The current version is provided on pdf for verification.

In figure 4, why you did not use a more closely related bacterium species, such as Serratia marcescens or some similar species as an outgroup in phylogenetic analysis?

Response: Serratia marcescens has been replaced as the outgroup, and topology of the tree confirms its position as basal of the analyzed strains. Figure 4 has been updated in the manuscript.

In the whole manuscript, there is a mixture of short and full names of the microorganisms, please review to make sure that the microorganism's names are in the full form the first time mentioned and then, only use the short name.

Response: Scientific names have been carefully reviewed and corrected when the genus was in full form twice or more throughout the text. Figures kept the full form for them to be self-explanatory.